# Somaclonal Variation—Advantage or Disadvantage in Micropropagation of the Medicinal Plants

**DOI:** 10.3390/ijms24010838

**Published:** 2023-01-03

**Authors:** Georgiana Duta-Cornescu, Nicoleta Constantin, Daniela-Maria Pojoga, Daniela Nicuta, Alexandra Simon-Gruita

**Affiliations:** 1Department of Genetics, Faculty of Biology, University of Bucharest, 06010 Bucuresti, Romania; 2Department of Biology, Ecology and Environment Protection, Faculty of Science, University “Vasile Alecsandri”, 600115 Bacau, Romania

**Keywords:** medicinal plants, in vitro micropropagation, somaclonal variation

## Abstract

Cell and tissue plant cultures are used either to save vulnerable species from extinction or to multiply valuable genotypes, or both, and are widely applied for economically important plant species. For medicinal plants, the use of in vitro technologies for the production of secondary metabolites and pathogen-free plants has been greatly developed. Two opposite aspects characterize the in vitro micropropagation of medicinal plants: maintaining genetic fidelity for the perpetuation and preservation of elites, and the identification and exploitation of somaclonal variations associated with new, useful traits. A balance between what is advantageous and what is undesirable is necessary, and this implies the identification of somaclonal variability at all levels, from the phenotypic to molecular ones. This review addresses the somaclonal variation arising from the in vitro multiplication of medicinal plants from three perspectives: cytogenetics, genetics, and epigenetics. The possible causes of the appearance of somaclones, the methods for their identification, and the extent to which they are desirable are presented comparatively for different plant species with therapeutic properties. The emphasis is on the subtle changes at the genetic and epigenetic level, as it results from the application of methods based on DNA markers.

## 1. Introduction

The use of medicinal plants is as old as mankind [1], some of the oldest records regarding the use of plants for treating diseases dates back 1000–5000 years: the Sumerian clay slab from Nagpur [5000 years old] consists of 12 recipes for herbal medicines and makes references to 250 different plants [1,2], while the Chinese book *Pen T’Sao*, which dates back approximatively 2500 years, describes 350 different drugs, some of them still being used even today [e.g., *Podophyllum*—known today as wild mandrake, or ground lemon, *Rheum palmatum* L.—rhubarb, *Camellia sinensis* (L.) Kuntze—tea plant, ginseng, jimson weed, cinnamon bark, and ephedra] [2,3]. From the ancient Greeks, we have numerous data regarding the use of plants in human medicine: Homer mentions in *The Iliad* and *The Odyssey* 63 plants used in pharmacotherapy [4]; Herodot (500 BC) endorses the health benefits of cabbage, mustard, hellebore, and garlic; Hippocrates describes in his works more than 300 medicinal plants [5]; while Dioscorides, who is considered the father of pharmacology, describes 657 drugs, from 944 different plants [6]. There are also records dating from ancient Rome. For example, Pliny the Elder (23 AD-79) documents over 1000 wild and cultivated medicinal plants, [7,8], while Galen (131 AD–200) describes in his book more than 220 plants, 61 minerals, and 19 animal drug products [9]. The turning point in the use of medicinal plants was at the beginning of the 19th century, when the advances in chemical methods allowed the discovery and isolation of numerous active compounds: alkaloids from poppy (1806), ipecacuanha (1817), strychnos (1817), quinine (1820), glycosides, tannins, saponosides, etheric oils, vitamins, hormones, etc. [10]. The 20th century brought with it the development of synthetic pharmaceutical products, which, for a while, reduced the use of natural medicines of vegetable origin. With the understanding that green medicines are healthier and safer, the interest in plant species with therapeutic properties has increased.

At present, there is a very high demand for a wide variety of medicinal plants on the market; therefore, these species face numerous challenges in their natural habitats, especially due to excessive harvesting and environmental pollution, which lead to a dramatic decrease in genetic diversity within these species. In vitro plant cultures are a more suitable alternative to solve these alarming problems. The in vitro multiplication of medicinal plants allows for obtaining disease-free plants on a large scale, as well as commercial-scale production of plant metabolites, thus compensating for habitat degradation and preserving natural plant populations. Besides, the in vitro culture of plant cells and tissues represents not only an effective method of plant regeneration but also a tool for manipulating the plant genome. The morphogenetic reaction and the evolution of explants in in vitro cultures can be different: the direct regeneration of new plants (through organogenesis or direct somatic embryogenesis), or the generation of an undifferentiated tissue called callus—an amorphous mass of parenchymal cells (of different size, shape, appearance, color, texture, proliferation rate, etc.) [11].

For medicinal plants, the use of in vitro cultivation technology has two main purposes: 1. to multiply some valuable genotypes, maintaining the integrity of their genetic profile, and 2. to obtain new somaclones with the highest possible content of bioactive compounds [12,13,14]. Therefore, it is necessary to rigorously monitor the genetic and epigenetic variation of the regenerants in order to maintain, on one hand, the genetic integrity with regard to the mother plant and, on the other hand, to capitalize on those useful randomly appear somaclonal variations which improve the therapeutic properties of medicinal plants, without additional costs.

One way to achieve this goal is the thorough analysis of culture-derived plants at all levels: chromosomal, DNA sequence, and epigenome, using modern methodologies based on DNA markers or genome-wide DNA screening.

## 2. In Vitro Propagation of Medicinal Plants

In vitro plant regeneration depends, to a large extent, on the genotype of the explant donor plant. Studying the influence of different explants, combinations of growth regulators, and various types of nutrient media on callogenesis and in vitro regeneration in six chamomile genotypes, Ahmad et al. (2021) [15] observed that regeneration was 77.5% genotype-dependent in direct embryogenesis and 77% in the case of indirect embryogenesis [15]. Sometimes, due to somatic mutations in the donor plant (that is, due to the chimeras in the explants), the appearance of somaclonal variations can be observed in the regenerants [16].

Highly differentiated tissues such as roots, leaves, and stems generally produce more somaclonal variations than explants with pre-existing meristems, such as axillary buds and shoot tips [17]. The data from the literature shows that, in medicinal plants, the most effective explants, which allows shoot regeneration directly or via a callus, are represented by leaf fragments, nodal and internodal stem fragments, and apical tips. Different types of explants react differently to the same nutritional variant. For example, the calli obtained by inoculating fragments of internodes, leaves, and roots from *Rhodiola rosea* L. on MS (Murashige Skoog) medium, supplemented with BAP and 2,4-D, were different in terms of color, proliferation speed, and organogenetic capacity [18].

An important factor for the success of in vitro multiplication is the basic formula of the nutrient medium and, for medicinal and aromatic plants, MS is the most common basic medium used in in vitro multiplication. The induction of callus and the regeneration of plants via callus depends on the presence, in the nutrient medium, of different types, concentrations, and combinations of phytohormones, especially auxins and cytokinins [19]. In general, it was found that the presence of auxins in the culture medium is necessary for the induction of calli, their type, and concentration, varying from one species to another [20,21], but the morphogenetic response of the explants also depends on the endogenous levels of phytohormones in the plants [22]. For example, the generation of an organogenic callus in *Mentha piperita* L. was possible when a leaf disc explant was cultured on MS medium containing 1.5 mg/L NAA+ 0.2 mg/L BAP, even if different auxins (NAA, 2,4-D and IAA) were tested in combination with different BAP concentrations (0.5–2.0 mg/L) [23,24].

For the development and evolution of plant explants, the cultures must be maintained under controlled conditions of light, temperature, photoperiod, and humidity. For example, lighting conditions can be stressful for inoculated explants, resulting in regenerants with abnormal phenotypes [25].

The age of the culture and the number of subcultures favor the appearance of somaclonal variations. In *Curcuma aromatica*, Mohanty et al. (2008) [26] obtained regenerants, and it was proven cytophotometrically that some of them had polyploid nuclei. The frequency of polyploid cells was higher in the case of regenerants obtained from a 180-day-old callus compared to those in a 6-day-old callus. So, the increased frequency of polyploid cells could be attributed to the prolonged callus culture [26].

## 3. Somaclonal Variation

The term ”somaclonal variation” was first introduced by Larkin and Scowcroft (1981) [27] to designate the genetic variation that occurs in plants regenerated from any type of cell culture. Currently, somaclonal variation means the variations that occur in clonally propagated plants from a single donor and represents a combination of morphological, cytological, biochemical, and genetic/epigenetic changes [16,28].

When talking about the establishment of a standardized micropropagation procedure and/or the establishment of commercial plantations for species with economic importance, somaclonal variability represents a big issue, because, in these cases, it is necessary to strictly maintain the characteristics of the individuals selected for the initiation of in vitro cultures. At the same time, somaclonal variability is a source of genetic diversity that breeders can use for the generation of new plants with superior economic traits [29,30,31,32]. Somaclonal variation may or may not be expressed at the phenotypic level and may have a genetic or epigenetic origin. The genetic changes are either at the chromosomal level, largely changes in chromosome number (aneuploidy or polyploidy) or structure (deletions, duplications, insertions, translocations), or at the level of the DNA sequence, mostly being point mutations in the DNA sequence. The prevailing epigenetic variations are gene amplifications and/or modifications of normal methylation patterns of DNA and histones [33]. Therefore, in order to identify the true-to-type replicants obtained after in vitro propagation of a genotype, it is essential to use methods for evaluating somaclonal variation [34]. The scientific community agrees and recommends the use of several different techniques to assess the phenotypic, cytological, phytochemical, and genetic profiles of the in vitro regenerated plants. [35].

## 4. Cytogenetic Analyses and Somaclonal Variation

The in vitro cultivation conditions affect mitotic stability, thus the variability in chromosome number and structure is more common in plant cell or tissue cultures than in the natural environment [36]. This cytogenetic variation appears to be more frequent in callus cultures than in other culture types, because callus formation implies a dedifferentiation phase followed by an uncontrolled cell division [37,38]. Nevertheless, genetic variation at the chromosomal level could be also noticed in plants regenerated from other in vitro cultivation methods [39,40] (Table 1).

Chromosome studies from plant tissue cultures are used to distinguish the cytological true-to–type regenerated plants from variants [41] and target the total number of the chromosomes, their relative size, centromere position, and, obviously, structural abnormalities.

For example, Al-Zahim et al. (1999) [42], in a study on garlic, analyzed a total of 75 somaclones regenerated from long-term callus cultures originating from three parental clones; they identified 16% abnormalities that included tetraploidy, aneuploidy (with either 31 or 15 chromosomes), and an extension of the secondary constriction. Gokhale et al. (2015) [43], in a study on *Oroxylum indicum* (L.) Vent., an endangered medicinal tree, showed that indirectly regenerated somaclones and callus cells exhibited substantial variations (25% in somaclones and 33% in callus), represented by tetraploidy and aneuploidy (with either the loss or gain of chromosomes), although cells with supernumerary chromosomes were also identified in the mother plant.

In a study made on a long-term citrus (*Citrus sinensis* L.) culture, Hao et al. (2002) [44] indicated that an embryogenic callus presented, in a small proportion, tetraploidy and aneuploidy (2.3%, i.e., 3/130, and 3.9%, i.e., 5/130, respectively) and some mitotic irregularities such as lagging chromosomes, chromosome bridges, unequal chromosome distribution, and meiosis-like division in anaphase and telophase cells. However, the cytological examination of somatic embryos and plants regenerated from the callus showed that they were diploid. A similar situation was noticed for *Lathyrus sativus* L., a medicinal plant used in the homeopathic treatment of spinal cord affections, 26% of the regenerants derived from long-term callus cultures frequently presented precocious and/or late separations of bivalents, anaphase laggards, anaphase bridges with or without acentric fragments, chromosome rings, and other common chromosomal rearrangements [45].

Using flow-cytometry, Mohanty et al. (2008) [26] detected that the callus (initiated from shoots)-regenerated plants of *Curcuma aromatica* Salisb. were mostly exclusively diploid (75.22%), and only 24% of them were partially diploid; the analysis of the distribution of DNA content confirmed the presence of diploid, tetraploid, and octoploid cells.

Another important medicinal plant is *Aloe vera* L., with many cultivars growing naturally or cultivated in different parts of the world. In a study made on natural and in vitro regenerated plants of sweet aloe, Das et al. (2010) [41] showed that the karyotypes of the cultivars of *A. vera* (L.) Burm.f., although very similar, can be distinguished, mainly based on the average chromosome length, type, and position of satellites and secondary constrictions, and that the regenerated plants were very stable genetically.

According to Radic et al. (2005) [46], *Centaurea ragusina* L. (a medicinal plant with antimicrobial properties) has a very stable genome, even if it is cultured for long periods of time. The authors investigated plants regenerated from 94 callus subcultures and observed that the maximum percentage of mitotic abnormalities was 12.4%, most of them being represented by irregular anaphases, twisted equatorial plates, and lagging chromosomes.

Although cytogenetic studies that investigate somaclonal variations in medicinal plants regenerated through in vitro cultures are rare, they all confirm that the highest percentage of chromosomal abnormalities is observed, as expected, in callus and protoplast cultures, with a predominance of polyploidy, followed by aneuploidy. One explanation is the fact that the regenerants are the result of a long series of mitoses during which numerous mutations appear spontaneously and uncontrolled, and the greater the number of subcultures, the more these mutations affect a greater amount of genetic material, as a result of the disruption of the cell division control mechanisms [29]. More than point mutations, chromosomal or genomic ones have phenotypic consequences, some of which are associated with economically/commercially useful characters. A summary of cytogenetic abnormalities observed in in vitro-regenerated medicinal plants is presented in Table 1.

## 5. Variation at DNA Level

In the case of in vitro-regenerated plants, not all the genetic variations that appear in the somaclones have a phenotypic expression, mainly because these modifications either are not in the coding sequences or do not alter the gene product to such an extent that can be observed in the phenotype [16].

To date, there are available numerous molecular techniques that can detect genetic variations between source plants and somaclones. Most of these techniques are based on different types of DNA markers, due to the fact that these markers are phenotypically neutral and are not influenced by any developmental changes or environmental factors [16]. The most used DNA markers are inter simple sequence repeats (ISSR), random amplified polymorphic DNA (RAPD), restriction fragment length polymorphism (RFLP), microsatellites (SSR), amplified fragment length polymorphism (AFLP), and start codon-targeted (SCoT) polymorphism [47,48,49,50,51].

**Table 1 ijms-24-00838-t001:** Cytogenetic abnormalities in micropropagated medicinal plants.

Plant Species	Type of In Vitro Culture	Cytogenetic Abnormality	Ref.
*Cuminum cyminum* L. (2n = 14)	cell suspension culture (CSC)	29% with 12 chromosomes, 15% tetraploid, 4% with 13 chromosomes, 1% with 27 chromosomes	[52]
	root tip cells of plants regenerated from CSC	chromosome no. ranged between 12 and 28	
*Plantago ovata* Forssk. (2n = 8)	Second-generation callus cell	numerical variation and other aberrations	[53]
	plant regenerated from second-generation callus	normal diploids	[54]
*Coffea arabica* L. (2n = 44)	plant regenerated from a 27-month-old cell culture	23–25% of the cell presented aneuploidy (2n − 1, − 2 or − 3)	[55]
*Cyphomandra betacea* (Cav.) Sendt. (2n = 24)	short-term (1 and 2 years) and long-term (7 and 10 years) calli	aneuploidy (43, 45, and 46 chromosomes) and tetraploidy	[56]
	plants regenerated from short-term embryogenic cultures (1 and 2 years)	normal (diploid)	
	plants regenerated from long-term embryogenic cultures (7 and 10 years)	tetraploid	
*Hypericum* perforatum L. (2n = 16)	plants regenerated by adventitious shoot formation	diploids (2n = 2x = 16), triploids (2n = 3x = 24), tetraploids (2n = 4x = 32), and mixoploids	[57]
*Withania somnifera* (L.) Dunal (2n = 48)	regenerated plants attained through indirect organogenesis from leaf explants	no modification of chromosome number and structure	[58]
*Carica papaya* L. (2n = 18)	somatic embryos	diploid (88%), tetraploid (6%), and aneuploid (6%) plantlets	[59]
*Dioscorea floribunda* Mart & Gall (2n = 36)	plants representing a single clone regenerated from stem tissue	diploids, mixoploids, and tetraploids	[60]
*Curcuma longa* L. (2n = 63)	root tips of callus-derived regenerants from the field	rare diploids with polymodal distribution of DNA content peaks	[61]
*Tylophora indica* R.Br. (2n = 22)	plants obtained by direct organogenesis from leaves	cytologically stable, no abnormality	[62]
*Lathyrus sativus* L. (2n = 14)	plants regenerated from long-term callus cultures	26% with one or more interchanges and/or loss of chromosome segments	[45]

Many researchers prefer to use a combination of different markers [63,64] since each of these approaches has advantages and limitations in the assessment of genetic variation [46]. For example, ISSR and RAPD have a major disadvantage because are dominant markers, i.e., do not allow discrimination between heterozygote and homozygote individuals. Moreover, numerous studies show that, when used alone, RAPD generated only monomorphic bands across all the analyzed plants (regenerants/mother plant) [65,66,67,68] and had a low degree of reproducibility.

Govinden-Soulange et al. (2010) [69] tested 30 different RAPD primers in order to assess the variation in micropropagated plantlets of *Hibiscus sabdariffa* L., and only 3 of them produced polymorphic bands, while Haque et al. (2017) [28], working on *Hibiscus cannabinus* L., screened 25 decametric primers and selected only 3 for the analysis of the whole sample set. However, the ISSR markers, applied comparatively between the plantlets and the mother plant revealed, in general, the clonal nature of the analyzed regenerants, highlighting the genetic fidelity of in vitro propagation, even after several years [70,71,72].

There are a number of studies in which the somaclonal variation analysis is carried out using a combination of two or three types of markers (Table 2), such as SCot, ISSR, and RAPD, and, in all cases, the research revealed the genetic fidelity of the regenerated plantlets (even for those that had passed through the callus stage) [73,74,75,76,77,78].

The use of AFLP markers is quite laborious, due to the fact that they require high-molecular-weight DNA, and, when the mother plant and micropropagated regenerants were compared, AFLP generated very low levels of polymorphism (Table 2). More often, the AFLP technique is used in combination with methylation-sensitive restriction enzymes in order to assess the epigenetic somaclonal variation associated with the modification of the whole genome methylation [79,80].

SSR markers have the advantage that they can differentiate between homo- and heterozygotes and do not require high-molecular-weight DNA (because the amplified sequences are between 100–300 pb), but they have the major disadvantage that they involve high costs if adequate primer sequences for the species of interest are unavailable. Moreover, SSR markers are prone to have “null alleles”, i.e., no amplification of the intended PCR product, due to the mutation that can occur in the primer annealing sites, which may lead to errors in scoring. SSR markers are considered genetically stable because, when used, they did not reveal any polymorphism, or only a very low level of it, between the original plants and the regenerants [81,82,83].

**Table 2 ijms-24-00838-t002:** Identification of somaclonal variations by DNA markers.

Plant Species	Type of Analysed Tissue	Molecular Markers	Results	Ref.
*Hibiscus sabdariffa* L.	single nodes explants, leaf from 10 regenerants + mother plant	RAPD (3 out of 30 were informative)	RAPD polymorphism between explants and mother plant	[69]
*Hibiscus cannabinus* L.	leaf tissue from 27 micropropagated plants	RAPD (3 out of 25 were informative)	- 68.18% polymorphism between mother plant and micropropagated plants - the decrease in NAA and the increase in BAP increases the percentage of polymorphism	[28]
*Silybum marianum* L.	callus tissues, leaves of regenerated plants, seed-derived plantlets, and plantlets	RAPD (9 out of 10 were informative)	OPC 10 revealed polymorphism all other 8 primers—monomorphic bands	[67]
*Chlorophytum borivilianum* Santapau & R.R.Fern.	leaves from 15 micropropagated plants and one field-grown plant	31 RAPD primers	100% monomorphism—all RAPD profile genetically similar to mother plant	[68]
*Humulus lupulus L.*	leaf tissue 10 explants/MS variant	16 RAPD primers	9.6% scoreable polymorphisms	[65]
*Celastrus* *paniculatus* Willd.	40 in vitro regenerated plantlets, rooted microshoots, acclimatized plantlets	RAPD (21 out of 30 were informative) + ISSR (12 out of 20 were informative)	100% monomorphism	[73]
*Pavetta indica* L.	leaf tissue	6 RAPD + 5 ISSR primers	100% monomorphic bands	[75]
*Thunbergia coccinea* Wall. ex D.Don	leaf tissue from mother plant, in vitro-raised direct regenerants, callus mediated plants	12 RAPD + 9 ISSR primers	Jaccard’s similarity coefficient 0.9542–1.000—all plants, even those that passed through the callus stage, proved to be genetically stable.	[74]
*Anoectochilus formosanus* Hayata	20 plantlets, sub-cultured in vitro every 3 months for a period of more than 5 years	ISSR (17 out of 50 were informative)	2.76% polymorphism—low risk of genetic instability, high genetic fidelity	[71]
*Plantago major* L.	callus samples from 18 in vitro-raised plants	ISSR (6 out of 18 were informative)	98.61% polymorphism	[84]
*Orthosiphon stamineus*	fresh leaf tissue from 10 in vitro regenerants after the 3rd subculture	ISSR (10 out of 20 were informative)	7.32% polymorphism	[70]
*Zingiber officinale* Roscoe.	leaf tissue + callus	4 ISSR primers	11.11%–42.86% polymorphism	[85]
*Salvia bulleyana* Diels.	4 shoot lines + 1 control	15 ISSR primers	not a significant somaclonal variations	[72]
*Cinchona officinalis* Diels.	leaf tissue + callus	ISSR (6 out of 13 were informative)	- 20–39% polymorphism - presence of 2,4-D correlated with somaclonal variation	[86]
*Pittosporum eriocarpum* Royle.	leaf tissue from 8 hardened plants randomly selected + mother plant	SCoT (10 out of 20 were informative) + ISSR (10 out of 15 were informative) + RAPD (10 out of 15 were informative)	97% similarity among micropropagated plants and mother plant	[78]
*Rauwolfia tetraphylla* L.	callus regenerants (4 from leaf + 3 from stem)	10 SCoT primers + 10 ISSR primers + 10 RAPD primers	absence of somaclonal variation in regenerants—100% monomorphic bands all 30 primers	[77]
*Dendrobium fimbriatum* Lindl.	leaf tissue from mother plant + plants regenerated on Mitra ± hormones	25 RAPD primers + 34 ISSR primers + 18 SCoT primers	100% monomorphism between plants regenerated on Mitra medium ± hormones	[76]
*Aerva lanata* (L.) Juss. ex Schult.	leaf tissue for 3 samples	5 combinations of 3 forward + 3 reverse SRAP primers	somaclonal variation in regenerants	[87]
	leaf tissue from mother plant + 9 randomly regenerants	10 RAPD primers	100% monomorphic bands	[66]
*Ducrosia anethifolia* (DC.) Boiss.	8 regenerated plants + mother plant	AFLP analysis—2 different digestion systems: *Mse*I/*EcoR*I and *Bgl*II/*Mse*I	- UPGMA cluster analysis revealed two main groups: mother plant/all the regenerants - all the identified variations from the regenerated plants result from genome methylation.	[79]
*Polyscias filicifolia* (C.Moore ex E.Fourn.) L.H.Bailey	leaf tissue from mother plant + 45 regenerants from each primary/secondary/tertiary somatic embryo	AFLP with 8 primers for MSeI/EcoR1	3.51% polymorphism between mother plant and regenerants	[80]
		metAFLP with 8 primers for KpnI/MseI + 8 primers for Acc65I/MseI	- sequence variation 1.25–0.75% - de novo methylation 0.41–0.18% - demethylation 0.54–0.84%	
*Parmentiera cereifera* Seem.	20 regenerants + mother plant.	SSR primers (36 out of 38 were informative)	micropropagated plants were genetically stable–4.49% polymorphism	[81]
*Lilium candidum* L.	leaf tissue from mother plant + regenerated bulbils + somatic embryos + acclimatized plantlets	12 SSR primers	no somaclonal variation after micropropagation—100% monomorphism	[82]
*Cannabis sativa* L.	leaf tissue from 9 micropropagated plants + donor plant	12 SSR primers	no somaclonal variation after micropropagation—100% monomorphism	[83]
*Withania somnifera* (L.) Dunal	mother plant + 10 micropropagated plantlets	12 SCoT primers	0.12% polimorphysm	[88]
		7 combinations of SRAP primers	- 0.0417 difference between mother plant and micropropagated plantlets - the phylogenetic analysis differentiates the plantlets from the control—2 different groups	
*Artemisia absinthium* L.	plant tissue in vitro *+* in vivo raised plants 10 replicates per treatment (MS + IBA)	ISSR primers (5 out of 15 were informative)	- 98% monomorphic bands for the indirect in vitro regenerants - 100% monomorphic bands for regenerants from nodal explants	[89]
		SSAP—*Mse*I enzyme + primers against LTR region and RNase H motif	- variability identified in callus derived plants - no somaclonal variation in plants directly regenerated from nodal explants	

ISSR = Inter Simple Sequence Repeats, RAPD = Random Amplified Polymorphic DNA, AFLP = Amplified Fragment Length Polymorphism, SSR = Simple Sequence Repeat, SCoT = Start Codon Targeted Polymorphism, SRAP = Sequence Related Amplified Polymorphism, SSAP = Sequence-Specific Amplification Polymorphism.

## 6. Somaclonal Epigenetic Variation

There is a growing number of studies indicating that the variability observed in plants regenerated through in vitro cultures is generated not only by genetic alteration, but also by epigenetic changes, represented by alteration of the global DNA methylation level, the chemical modifications of histones, which, in turn, are associated with shifts at the chromatin level (transition heterochromatin—euchromatin state), and also by de novo methylation processes, in which the RdDM pathway is essential. For example, it is well known that modifications such as the ubiquitination of H2BK143, mono-, di-, and trimethylation of H3K4, H3K36, and H3 and H4 lysine acetylation are associated, in *Arabidopsis* genome, with gene activation, while the mono-, di- and trimethylation of H3H9, H3K27, and H4K20, and the dimethylation of H4R3, are associated with gene inactivation [90,91,92,93]). Incorporation of different histone variants can contribute to chromatin remodeling, as shown by Sura et. al, 2017 [94] (incorporation of histone variant H2A.Z has a repressive role in transcription and counteracts unwanted expression in noninductive conditions).

This epigenetic variability, which frequently is observed at phenotype level, reflects, in reality, the adaptations of the cellular processes to the various constraints of the in vitro cultivation technology, that usually implies sequential dedifferentiation (formation of callus) and re-differentiation (regeneration of plants) [91,95,96,97,98,99,100]. The level of genetic instability and, hence, the epigenetic modification of somaclone genomes, and the mechanisms involved in these processes varies largely between plants species.

The most evaluated epigenetic somaclonal variation is **global DNA methylation**, which, in plants, is at higher levels compared with other eukaryotes, and is correlated with the expression of different genes, and changes in the mobility of the numerous transposable elements presented in plant genomes [99,101,102,103,104,105,106,107,108,109].

Over time, several methods have been applied to detect the level of methylation in in vitro-cultivated plants, but the most frequently used are MSAP (methylation-sensitive amplification length polymorphism), HPLC (high-performance liquid chromatography), HPCE (high-performance capillary electrophoresis), and, rarely, in the early period of these studies, MS-RFLP (methylation-sensitive restriction fragment length polymorphism) [91].

The variation in the methylation pattern of the in vitro regenerated somaclones is a result of an admixture of factors, including the plant species, regeneration system, the epigenetic pattern in donor plants, length of cultivation period, composition of culture media, etc. [101]. For example, many in vitro micropropagation strategies imply direct somatic embryogenesis (SE), and there are studies indicating that this process is characterized by either hypermethylation or hypomethylation of DNA [110]. Usually, hypermethylation is associated with the promotion of SE and silencing of repetitive elements, while hypomethylation suppresses SE, but these processes are species-specific. For example, Fraga et al. (2012) [111], in a study on *Acca sellowiana* O.Berg., (a medicinal plant used to treat diarrhea, tumors, and microbial infections [112]) showed that a combined treatment with 5-azacytidine (AzaC) and 2,4-dichlorophenoxyacetic acid (2,4-D) determined an overall increase in the methylation level, corelated with a positive influence on SE induction, even if the response to the combined treatment (pulse 200 µM 2,4 D for 1 h + 50 µM AzaC in the culture medium) varied according to the individual genotype. The zygotic embryos of one accession presented an increase in the methylation level after 10 days of inoculation on the transformation media, but this level decreased after 20 days and increased again after 30 days, while, for the other analyzed accessions, a gradual increase in the methylation level was registered during the three checkpoints (10, 20, and 30 days after inoculation). [111]. Similar observations were made by Nic-Can et al. (2013) [113] on *Coffea canephora* Pierre ex A.Froehner, a plant rich in antioxidants and other chemicals, used to reduce the risk of diabetes mellitus, arterial hypertension, cardiovascular diseases, obesity, and even depression [114]. Contrariwise, Chakrabarty et al., (2003) [115] showed that, in Siberian ginseng, the induction of SE is associated with the hypomethylation of DNA, while hypermethylation is associated with the repression of it. Using two different approaches (HPLC quantification of nucleosides and MSAP technique), the researchers investigated two calli of the same genotype, one that formed embryos and one that did not, raised from the same culture conditions, and identified that 16.99% of 5′-CCGG-3′ sites in the genome of the non-embryogenic callus were cytosine methylated, while only 11.20% in case of the embryogenic callus tissue.

Regarding the **variation of DNA methylation level** in somaclones, one of the most interesting studies is that of Jaligot et al. (2000) [116], which investigated DNA methylation levels in two different palm calli: fast growing (that generate almost 100% of ”mantled” plantlets—with abnormalities in their floral development, involving an apparent feminization of male parts in the flowers of both sexes), and nodular compact calli (that generate only 5% abnormal plants). Using two different analysis methods (high-performance liquid chromatography—HPLC, and SssI-mediated reverse-dosage of CG-specific methylation) for assessing the methylation rates, the research team determined that there is a positive correlation between genomic DNA hypomethylation and abnormal in vitro clones regenerated from the fast growing calli. Contrary results were obtained for *Gentiana pannonica* Scop. by Fiuk et al., (2010) [117], which detected, using reversed-phase, high-pressure liquid chromatography and the metAFLP procedure, higher levels of methylation in regenerated plants than those registered as controls. In addition, a more in-depth analysis of the differences between the in vitro somatic embryo-derived plantlets and the mother plant, with quantitative molecular markers, revealed numerous tissue culture-induced variations, including sequence variations, changes in methylation patterns, and combinations of genetic and epigenetic changes, with the total level of culture-induced variation being ca. 16%.

There are also medicinal plant species in which no modification in the methylation profile was observed in the in vitro regenerated plants. For example, Gillis et al. (2007) [118] mass-propagated mature bamboos (*Bambusa balcooa* Roxburgh), via SE, using pseudospikelets, and didn’t detect, by methylation-sensitive AFLP (MSAP), any epigenetic differences between the regenerants and the mother plant. Similar results were obtained for *Myrtus communis* L. [119], in which the amount of 5-methylcytosine in the genomic DNA was studied by reversed-phase HPLC analysis of 2”-deoxynucleosides, both in shoots grown in the field and in shoots obtained in vitro, and subsequently rooted and acclimatized.

The regenerants obtained via callus culture are frequently associated with de novo DNA methylation/demethylation as well as histone modification [117]. In 2021, Zhang [96] investigated the epigenetic variation in *Populus nigra* L., regenerated from callus cells, analyzing the DNA methylation patterns of 5′-CCGG-3′ sites of five successive generations, and evaluated also their growth performance and physiological traits. The results indicated that all the regenerated plants had significantly lower levels of DNA methylation, with considerable differences between the first generation regenerated plants (G1) and those of the other four generations (G2–G5), but with no notable differences between the G3, G4, and G5 in vitro clones. The conclusion of the study was that the methylation pattern decreased mainly in the first and second generations, and became stable throughout subsequent generations.

Duarte-Aké et al., (2016) [120] investigated the changes in global DNA methylation levels in *Agave angustifolia* Haw. variant somaclones that, during the in vitro micropropagation process, differentiated in three different phenotypes: green (G), variegated (V), and albino (A). An analysis of the global DNA methylation level showed that the G variant had, during the first two subculture passages, a higher methylation level than the parent plants, but this level decreased during the next subcultures. The changes in methylation level were correlated with the appearance of V shoots in the G plantlets and A shoots in the V plantlets. The researcher concluded that there might be an “epigenetic stress memory” developed during in vitro micropropagation, which could cause a chromatin shift and the appearance of the other two variants, V and A.

Another medicinal plant investigated for its epigenetic stability after in vitro multiplication is hop, *Humulus lupulus* L. Peredo et al. (2006) [121] identified, using the MSAP technique, that the plants recovered from the first and second callus subcultures had very similar methylation profiles to the in vitro control pools, while the regenerants from the third callus subculture showed the highest genetic distance from the controls. Following this research, Peredo et al. (2009) [122] tested two hop accessions that were micropropagated for 2 years, and did not identify any genetic variation between the 12 cycles of micropropagated accessions and controls, but observed epigenetic variations when the plants grown from seeds and in vitro samples were compared.

Xu et al. (2004) [123] analyzed the DNA-methylation profiles of *Rosa hybrida* L. plants regenerated through different techniques: in vivo-grown greenhouse plants, in vitro-grown proliferating shoots at different passages, regenerants of embryogenic callus, regenerants of organogenic callus, as well as calli from undifferentiated callus (UC), embryogenic callus, and organogenic callus. Using an AFLP-based DNA-methylation technique, they identified that there is a great deal of variation in the methylation profiles between plants obtained via SE and in vitro organogenesis, mostly due to a demethylation of the second C from the 5′-CCGG-3′ sequence in somatic embryos.

Epigenetic variations in in vitro-regenerated plants, related to **histone modifications,** have been largely reviewed for model plants, such as *Arabidopsis*, rice, and maize [91,98,110,124], but not so much for medicinal plants. In a study on two *Agave* species (plants from this genus are used for wound-healing, treatment of diarrhea, dysentery, etc. [125]), de-la-Pena et al. (2012) [126] analyzed whether the adaptation to different in vitro culture systems (Magenta boxes—M and Bioreactors—B) epigenetically modifies different clones of *Agave fourcroydes* and *A. angustifolia,* and whether these epigenetic changes affect the regulatory expression of KNOTTED1-like HOMEOBOX (KNOX) transcription factors. For that, clones of both species were first in vitro-cultivated for 16 weeks, transferred for another 5 weeks into the two in vitro systems, and then transplanted ex vivo and raised for 2 months. A global DNA methylation level analysis revealed no significant differences between the *A. fourcroydes* clones grown in either M or B in vitro systems, but, when the two agave species were compared, there was a two-fold difference between them, independent of the in vitro system used. Moreover, the Western blot analysis using antibodies against the H3K4 di- and tri-methylated isoforms, as well as for the H3K9me2 and H3K36me2 markers, showed important changes in the histone methylation patterns between the in vitro and ex vitro clones. For example, the repressive H3K9me2 and the activation H3K36me2 markers accumulated in the in vitro samples but were absent/or in low amounts in the ex vitro ones. Furthermore, the research team showed that the expression of AtqKNOX1 and AtqKNOX2 genes was affected by the in vitro cultivation conditions in both *Agave* species and, using chromatin immunoprecipitation (ChIP), determined that the H3K4me3 and H3K9me2 markers were affected in the AtqKNOX1gene, in comparison with the AtqKNOX2 gene.

The epigenetic instability of an in vitro cultured plant is associated not only with the expression of histone genes, and the genes involved in post-translational modifications and chromatin regulation, but also with the misregulation of the **genes for non-coding RNA epigenetic systems.** As proven in the model species *Arabidopsis thaliana*, microRNAs (miRNA) and small interfering RNAs (siRNA) are essential regulators of plant development, guide DNA methylation in plants (especially de novo methylation), and act as transcriptional repressors [127,128]. miRNAs (usually 21–22 nt) mediate posttranscriptional gene silencing by mRNA degradation or by a repression of translation. siRNAs (24–26 nt) target specific DNA and histone sequences for methylation and heterochromatinization via a specialized RNA-dependent DNA methylation (RdDM) pathway [129].

Although progress has been made regarding the role of small RNA families in epigenetic processes, little is known about their response to in vitro cultivation conditions, and even less about their involvement in the occurrence of somaclonal variations.

In in vitro-grown strawberries (*Fragaria X ananassa*, Duch), microRNA genes changed their expression levels [130]. It must be noted that, even if strawberry is not a true medicinal species, in vitro studies showed that it contains a large amount of phenolic compounds with antioxidant and anti-inflammatory action, and has antimicrobial, anti-allergy, and anti-hypertensive properties [131]. Li et al. (2012) [132] investigated, by microarray, the differences in miRNA expression between conventional and in vitro micropropagated strawberry plants. They found that four miRNAs were differentially expressed between the two groups of plants: miR535 and miR390 were up-regulated, and miR169a and miR169d down-regulated in in vitro micropropagated plants. The changes in the levels of miR169a and miR390 were associated with altered phenotypes in tissue-cultured strawberry plants (altered stomatal aperture size and timing of flower bud differentiation) [130]. Another study on the strawberry plant identified a high expression of miR156, which lead to the assumption that miR156 is involved in rejuvenation in micropropagated plants [132]. Thus, it is obvious that even subtle changes in microRNA levels can have downstream effects on the expression levels of numerous genes [133].

The epigenetic pathway play an important role in protection against **transposable elements (TE)**, affecting their movement and expression. These mechanisms include transcriptional silencing via DNA methylation, histone modification, and alterations in chromatin packing, as well as RNA-directed DNA methylation (RdDM) [134].

Some TEs become reactivated in in vitro cultures, with epigenetic changes that occur through the cultivation process being responsible for their activation. Tanurdzic et al. (2008) [135] studied the epigenomic consequences of long-term culture on *A. thaliana* cells and identified that TEs with the strongest reactivation were *Athila* retrotransposons, and some members of the *Helitron, Mu, Vandal*, and *En/Spm* families of transposons. The activation of these TEs was correlated with an increase of 21-nt siRNA in the cultured cells. In contrast, TEs that remained silenced, such as the retrotransposons *Atlantys* and *AtGP*, generated 24-nt siRNAs at the same level as the mother’s vegetative tissue. The researchers concluded that euchromatinization in the in vitro cultivated cells of heterochromatin regions is associated with a change in the relative abundance of 21-nt siRNAs.

So far, in vitro culture-induced TE activity has been reported in various commercial plant species such as rye [136], maize [137] and rice [138], but, for medicinal plants, there are no results. Since the activation of TE can be a source of mutations, duplications, rearrangements, modification of gene expression, and, finally, genomic instability, it is necessary to deepen the studies regarding the involvement of TE in the generation of somaclonal variations in medicinal plants micropropagated in vitro.

## 7. Practical Consequences of Somaclonal Variation

In vitro micropropagation of medicinal plants represents an alternative to conventional crop improvement, ensuring the rapid propagation of disease-free plants, preservation of valuable genotypes, exploitation of somaclonal variations, and manipulation of plant genomes through genetic engineering. The production of secondary metabolites with therapeutic properties, useful in the pharmaceutical industry, is of major interest in the case of these plant species.

In medicinal plants, and not only those obtained by in vitro cultivation technologies, changes in the secondary metabolite synthesis patterns can be observed. These changes are most often generated by the genetic and epigenetic adaptations of the mechanisms of secondary metabolite production to the stressful conditions associated with the in vitro multiplication process.

From an economic point of view, somaclones provide a source of genetic variability that can be exploited to improve the biochemical properties of medicinal plants. For example, Roopadarshini and Gayatri 2012 [139] observed, in different somaclonal variants of *Curcuma longa* L., significantly higher amounts of curcumin, oleoresin, and volatile oil contents, than in control plants. Other research, conducted on *Viscum album* L. (a medicinal plant with anticancer properties), revealed variations in protein concentrations between different callus tissues regenerated from long stem segments. The authors assayed 100 individual callus tissues and, in approximately 10% of the calli, determined that the protein concentration was six-times higher than in the donor stem tissues. However, the protein content and its profiling were not identified [140]. Shyam et al., 2021 [141] regenerated plantlets from callus cells from two different *Brassica juncea* cv. Prakash. genotypes (a medicinal plant with many purposes: the seeds are used as treatment for tumors, abscesses, colds, lumbago, rheumatism, and stomach disorders, the roots to increase the milk supply in lactating women, while the oil is used as a treatment for skin disorders and ulcers [142]). They compared erucic acid, a fatty acid present in high concentrations (35–50%) in mustard oil, making it nutritionally unfavorable for the human diet, and other fatty acid contents in plantlets directly regenerated from cultured tissues (R_0_), their first-generation offspring (R_1_), and the parental plants. For one of the genotypes, the R_0_ and R_1_ somaclones presented much lower erucic acid content (5.48% and 5.52%, respectively) than the mother plant (41.36%), and higher amounts of palmitic and linolenic acids. In the R_0_ and R_1_ putative somaclone of the second genotype, no erucic acid was detected, despite its presence (1.07%) in the mother plant. Radomir et al., 2022 [143], compared a number of biochemical indicators in in vitro- and seed-generated *Mentha piperita* L. plants and identified some differences between the two types of regenerants: the content of photosynthetic pigments in the plants regenerated in vitro was higher than in the ones obtained from seeds, while the total phenol content in the plants obtained by classical methods was higher than in the in vitro ones. Shelepova et al., 2021 [144], investigated the essential oil (EO) profiles of field-acclimatized in vitro-micropropagated plants from three *Mentha × piperita* L. cultivars. Using gas chromatography–mass spectrometry (GC–MS) they pointed out that the main component of the EO in the control plants was menthol, while, in the one-year-old, field-acclimated plants, the in vitro regenerants were pulegone and menthone. However, in the second year of vegetation, the main EO components in field-acclimated peppermint plants were approximately the same as in the control plants.

In general, for true medicinal plants, there are only a handful of studies that correlate the genetic and epigenetic changes with gene expression, and analyze the phenotypic differences (including, here, the medically important phytoconstituent profiles) between donor plants and their field/greenhouse acclimatized in vitro clones. However, there are numerous studies on non-medicinal plants that highlight the phenotypic differences between the somaclones and the mother plant. For example, Hariedy et al., 2019 [145], obtained, by in vitro regeneration via callus, *Pelargonium graveolens* L’Herit somaclones with superior volatile oil (citronellol and geraniol) contents than the mother plant; in 2015, Dey et al. [64] identified *Cymbopogon winterianus* somaclones with high agronomic characters (plant height, diameter of bush, number of tiller/clump, number of leaves/clump, leaf length, leaf breadth, weight of 100 leaves, and citronellal and geraniol essential oil content). In the case of sugarcane (*Accharum officinarum* L.), Ahmed et al. (2019) [146] identified somaclones resistant to red rot, while Abo-Elwafa (2021) [147] identified somaclones that had higher significant values than the donor plant in terms of stalk height, weight, and number/fed and cane yield.

## 8. Conclusions

The genetic and epigenetic diversity of in vitro-obtained somaclones can be either advantageous or harmful, depending on the purpose for which the micropropagation is carried out. Maintaining genetic fidelity is essential because the aim is to preserve a useful character of economic importance, so the genetic or epigenetic variations reflected in the phenotype represent a big problem. However, it is necessary that the morphological, biochemical, genetic, or epigenetic differences between donor plants and clones be identified as early as possible, using a complex of screening techniques, especially since not all somaclonal variations at the level of the genome and epigenome are manifested phenotypically.

To resume, some of the advantages of producing somaclonal variations in medicinal plants are: 1. It represents a simpler, faster, and cheaper way to obtain new varieties of plants compared to classic breeding methods. Thus, it is also very useful for perennial species, with a long vegetation period; 2. The variations occur with high frequency, which is advantageous compared to conventional mutagenesis. 3. It is appropriate for plant species with limited genetic diversity. However, the generation of somaclonal variations does not require knowledge about the genome of the respective species. 4. It can introduce new traits, such as resistance to a spectrum of diseases, pathotoxins, herbicides, biotic and abiotic stress. 5. It can create varieties with an increased production of valuable metabolites with phototherapeutic properties.

The appearance of somaclonal variations in somaclones has also some major disadvantages, such as: 1. the variations are random and cannot be predicted; 2. sometimes they are unstable and nonheritable; 3. They can be associated with deleterious features, such as reduced fertility and growth rates, or low or absent powers of regeneration. 4. To confirm the stability of a cell/plant line obtained from somaclones, repeated selection is required.

In the micropropagation of medicinal plants, it is necessary to maintain a balance between what is advantageous and what is undesirable, and this implies the deepening of the studies on the origin and molecular mechanisms (genetic and epigenetic) of somaclonal variations, as well as the development of effective strategies for the analysis of this diversity. The epigenetic mechanisms underlying somaclonal variations must also be deciphered in medicinal plant species, especially the role of small RNAs in chromatin changes, gene silencing, and TE activation.

In addition, for the proper use of somaclonal variations, appropriate plant regeneration systems for medicinal plant species are a prerequisite for future genetic transformation, conservation, and breeding programs.

## Data Availability

Not applicable.

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
