# Peer review of "Somaclonal Variation—Advantage or Disadvantage in Micropropagation of the Medicinal Plants"

_ijms, 2023, doi:10.3390/ijms24010838_

Round 1
Reviewer 1 Report
The review entitled Somaclonal variation – advantage or disadvantage in micropropagation of the medicinal plants. The content of the review does not match the title of the manuscript, it does not discuss advantages or disadvantages of Somaclonal Variation in micropropagation of the medicinal plants. On the other hand, I find inconsistencies in what is said in the work compared to the references used.
Some examples:
A large part of the section dedicated to epigenetic changes refers to regulation of morphogenesis, not to somaclonal variation. In some cases claims are made that are not the results obtained in the cited reference.
Lane 293 A study on Acca sellowiana O.Berg., (a medicinal plant used to treat diarrhea, tu-mors and microbial infections [114]) showed that treatments with 5-azacytidine (AzaC) and 2,4-dichlorophenoxyacetic acid (2,4-D) increased the overall methylation level of in 295 vitro regenerants and, in this way, promoted somatic embryogenesis [115].
The paper says: Treatment with AzaC and 2,4-D-free resulted in a marked decrease in methylation for both accessions, ranging from 37.6 to 20.8 %. In treatment with 2,4-D and AzaC combined, the 85 accession showed increasing global methylation levels.
Lane 288 For example, many in vitro micropropagation strategies imply direct somatic embryogenesis and there are studies indicating that this process is characterized by both hypermethylation and hypomethylation of DNA [113]. The reference is a revision.
Lane 301 Jaligot et. al. (2000) [118] showed that hypomethylation is associated with the suppression of somatic embryogenesis in in vitro micropropagation of Elaeis guineensis Jacq. Using high performance liquid chromatography (HPLC) for assessing the methylation rates, the research team indicates that, not only in vitro proliferation induces DNA 304 hypermethylation in a time-dependent fashion but also that the loss of genomic methyl tion during the proliferation of a clonal line hindered it to generate somatic embryos.
The cited paper says in its conclusions: In the present study, we have established a relationship between genomic DNA hypomethylation and mantled somaclonal variation in oil palm, on two different plant materials (calli and leaves) originating from various genotypes and following two complementary approaches (HPLC quantification of nucleosides and SssI-mediated reverse-dosage of CG-specific methylation).
I reflect only some of the inconsistencies of the work to justify why, in my opinion, it should be rejected
Author Response
Dear reviewer,
Thank you very much for the review. We have carefully analyzed and considered all your observations and we have redone, we hope, the article so that it addresses all your observations.
Observation 1. ”The content of the review does not match the title of the manuscript, it does not discuss advantages or disadvantages of Somaclonal Variation in micropropagation of the medicinal plants.” - Response: We highlighted now the advantages and disadvantages of somaclonal variability in micropropagated medicinal plants in: chapter 1. - Introduction (lane 54-75), chapter 7 – Practical consequences of somaclonal variation (line 940 – 971, 977-980) and chapter 8 – Conclusions.
Observation 2. ”A large part of the section dedicated to epigenetic changes refers to regulation of morphogenesis, not to somaclonal variation.” - Response: We discuss the corelation between DNA methylation level and morphogenesis only for a copuple of species (Acca selowiana - 512, Coffea canephora – 522, siberian ginseng – 525) but in the remaining part of the chapter (533 – 768) we present data regarding the variation of DNA methylation level in somalones (for example in palm – 533, bamboo – 550, Myrtus communis - 728, pupulus nigra – 733, etc).
Observation 3: ”Lane 293 A study on Acca sellowiana O.Berg., (a medicinal plant used to treat diarrhea, tu-mors and microbial infections [114]) showed that treatments with 5-azacytidine (AzaC) and 2,4-dichlorophenoxyacetic acid (2,4-D) increased the overall methylation level of in 295 vitro regenerants and, in this way, promoted somatic embryogenesis [115].The paper says: Treatment with AzaC and 2,4-D-free resulted in a marked decrease in methylation for both accessions, ranging from 37.6 to 20.8 %. In treatment with 2,4-D and AzaC combined, the 85 accession showed increasing global methylation levels”. - Response: We have rephrased our discussions about this paper – 511 – 521.
Observation 4: ”Lane 288 For example, many in vitro micropropagation strategies imply direct somatic embryogenesis and there are studies indicating that this process is characterized by both hypermethylation and hypomethylation of DNA [113]. The reference is a revision.” – Response: we do not understand why the fact that the cited paper is a review is a problem, but we decided to erase altogether the phrase in question.
Observation 5: ”Lane 301 Jaligot et. al. (2000) [118] showed that hypomethylation is associated with the suppression of somatic embryogenesis in in vitro micropropagation of Elaeis guineensis Jacq. Using high performance liquid chromatography (HPLC) for assessing the methylation rates, the research team indicates that, not only in vitro proliferation induces DNA 304 hypermethylation in a time-dependent fashion but also that the loss of genomic methyl tion during the proliferation of a clonal line hindered it to generate somatic embryos. The cited paper says in its conclusions: In the present study, we have established a relationship between genomic DNA hypomethylation and mantled somaclonal variation in oil palm, on two different plant materials (calli and leaves) originating from various genotypes and following two complementary approaches (HPLC quantification of nucleosides and SssI-mediated reverse-dosage of CG-specific methylation).” – Response: We have rephrased our discussions about this paper – 533 – 549.
We hope that after these changes you will reconsider your opinion regarding the rejection of this paper.
Reviewer 2 Report
The manuscript entitled "Somaclonal variation – advantage or disadvantage in micropropagation of the medicinal plants" present a review about genetic cytogenetic and epigenetic changes from micropropagation of the medicinal plants.
I found the mauscript well written, providing also Tables which contain all the significant information about previous studies that have been conducted in this field.
Author Response
Dear reviewer,
Thank you very much for the review and we glad that you liked our paper.
Reviewer 3 Report
The authors reviewed a fairly large number of publications related to the problem of somaclonal variation in medicinal plants. In general, in this volume of text, three genetic bases for the appearance of somaclonal variation are sufficiently reflected. However, in my opinion, one more section is missing, where the practical consequences of somaclonal variation would be described. That is, its connection with a change in the properties of the final product in medicinal plants. And such a final product is a variety of secondary metabolites, exactly what medicinal plants are grown for. Otherwise, it is not clear why the authors consider somaclonal variation at all, if it does not affect the production of medicinal compounds in any way. Such a section should be inserted at number 5, immediately after section 4. A couple of paragraphs and a few facts will suffice. This will justify the need to study somaclonal variation.
The «Brief history of the use of medicinal plants» section is oversized. Maybe it should be removed altogether?
The last paragraph before the Conclusion section. It is desirable to consider the following question. All medicinal plants propagated in vitro, one way or another, are ultimately planted in open ground. Somaclonal variations can accumulate in these plants during in vitro cultivation. How might these plants be different from plants that have not been cycled in vitro? For example, this is considered in the works: 1) Catalano, C., Carra, A., Carimi, F., Motisi, A., Abbate, L., Sarno, M., Carrubba, A. Long-Term Field Evaluation of Conventional vs. Micropropagated Plants of Chrysanthemum cinerariifolium. Agronomy, 2022, 12(11), 2756.
2) Shelepova, O. V., Dilovarova, T. A., Gulevich, A. A., Olekhnovich, L. S., Shirokova, A. V., Ushakova, I. T., Baranova, E. N. Chemical Components and Biological Activities of Essential Oils of Mentha × piperita L. from Field-Grown and Field-Acclimated after In Vitro Propagation Plants. Agronomy, 2021, 11(11), 2314.
3) Radomir, A. M., Stan, R., Pandelea, M. L., Vizitiu, D. E. In vitro multiplication of Mentha piperita L. and comparative evaluation of some biochemical compounds in plants regenerated by micropropagation and conventional method. ACTA SCIENTIARUM POLONORUM-HORTORUM CULTUS, 2022, 21(4), 45-52.
In these studies, differences were assessed either by the DNA content or by the chemical composition of secondary metabolites. You can devote at least 1-2 paragraphs to such a discussion.
Minor remark
Line 118-123 - It matters from which plant organ the explant was taken. It has been shown that even in the same plant, cells from different organs and tissues can differ greatly in ploidy.
This manuscript requires a major revision
Author Response
Dear reviewer,
Thank you very much for the review. We have carefully analyzed and considered all your observations and we have redone, we hope, the article so that it addresses all your observations.
Observation 1: ” However, in my opinion, one more section is missing, where the practical consequences of somaclonal variation would be described. That is, its connection with a change in the properties of the final product in medicinal plants. And such a final product is a variety of secondary metabolites, exactly what medicinal plants are grown for.” - Response: We have introduced a new chapter, (7 – Practical consequences of somaclonal variation) – where we discuss your suggestions (928 – 986).
Observation 2: ”Line 118-123 - It matters from which plant organ the explant was taken. It has been shown that even in the same plant, cells from different organs and tissues can differ greatly in ploidy” - Response: We have rephrased our discussions – 295 – 298
Observation 3: ” The «Brief history of the use of medicinal plants» section is oversized. Maybe it should be removed altogether?” - Response: we have downsized this part and introduced it to chapter 1 (Introduction – 28-52), so the Chapter ”Brief history of the use of medicinal plants” have been removed
We hope that these changes are in agreement with what you are looking for in this paper.
Reviewer 4 Report
In this manuscript, Georgiana et al. have reviewed the somaclonal variation arising from the in vitro multiplication of medicinal plants from three perspectives: cytogenetics, genetics, and epigenetics. The possible causes of the appearance of somaclones, the methods for their identification, and the extent to which they are desirable are presented comparatively for different plant species with therapeutic properties. The emphasis is on the subtle changes at the genetic and epigenetic level, as it results from the application of methods based on DNA markers. Although the topic is attractive there are some concerns that should be addressed.
-Generally, the manuscript is well organized but there are some typographical and grammatical errors.
-The paper title is well stated; it is informative and concise.
-Abstract is well structured.
-The introduction was not well written, and it is too briefly presenting the subject.
-7. Somaclonal epigenetic variation: It should be presented based on three subtitles: DNA methylation, Histone modifications, and RNA interference (small RNAs).
-Conclusion should be improved. I recommend discussing the application of machine learning (https://doi.org/10.1007/s00253-022-11963-6; https://doi.org/10.1007/s00253-020-10888-2) for a better understanding of somaclonal variation in the conclusion part.
Author Response
Dear reviewer,
Thank you very much for the review. We have carefully analyzed and considered all your observations and we have redone, we hope, the article so that it addresses all your observations.
Observation 1: ” The introduction was not well written, and it is too briefly presenting the subject.” – Response – we have rewritten the entire” Introduction” (line 28 – 79).
Observation 2: ”Somaclonal epigenetic variation: It should be presented based on three subtitles: DNA methylation, Histone modifications, and RNA interference (small RNAs).” – Response – even if data regarding the somaclonal variation, histone modifications and small RNA are scarce for medicinal plants species we have addressed these aspects from the available data on other plant species (mostly model ones, like Arabidopsis, rye, maze, etc.) – lane 769 - 927
We hope that these changes are in agreement with what you are looking for in this paper.
Round 2
Reviewer 3 Report
The authors have done a thorough work of making corrections to the text. Therefore, I believe that the manuscript can be accepted for publication in the IJMS in its present form.
For the future for authors. It is very inconvenient to read your manuscript with corrections. It is better to provide new text, in which the introduced or corrected fragments can be highlighted, for example, in yellow.
Reviewer 4 Report
All my comments have been addressed.